# Utilization of Cumbeba (*Tacinga inamoena*) Residue: Drying Kinetics and Effect of Process Conditions on Antioxidant Bioactive Compounds

**DOI:** 10.3390/foods10040788

**Published:** 2021-04-06

**Authors:** João Paulo de Lima Ferreira, Alexandre José de Melo Queiroz, Rossana Maria Feitosa de Figueirêdo, Wilton Pereira da Silva, Josivanda Palmeira Gomes, Dyego da Costa Santos, Hanndson Araujo Silva, Ana Paula Trindade Rocha, Anna Catarina Costa de Paiva, Alan Del Carlos Gomes Chaves, Antônio Gilson Barbosa de Lima, Romário Oliveira de Andrade

**Affiliations:** 1Department of Agricultural Engineering, Federal University of Campina Grande, Campina Grande 58429-900, Brazil; joaop_l@hotmail.com (J.P.d.L.F.); rossanamff@gmail.com (R.M.F.d.F.); josivanda@gmail.com (J.P.G.); 2Department of Physics, Federal University of Campina Grande, Campina Grande 58429-900, Brazil; wiltonps@uol.com.br; 3Department of Technology in Agroindustry, Federal Institute of Acre, Xapuri 69930-000, Brazil; dyego.csantos@gmail.com; 4Department of Processes Engineering, Federal University of Campina Grande, Campina Grande 58429-900, Brazil; hanndson@gmail.com; 5Department of Food Engineering, Federal University of Campina Grande, Campina Grande 58429-900, Brazil; ana_trindade@yahoo.com.br (A.P.T.R.); ann.paiva@hotmail.com (A.C.C.d.P.); 6Department of Mechanical Engineering, Federal University of Campina Grande, Campina Grande 58429-900, Brazil; alandcgc@hotmail.com (A.D.C.G.C.); antonio.gilson@ufcg.edu.br (A.G.B.d.L.); 7Department of Technology in Agroindustry, Federal Institute of Alagoas, Piranhas 57460-000, Brazil; romario.andrade@ifal.com.br

**Keywords:** by-product, pretreatments, convective drying, effective water diffusivity, phenolic compounds

## Abstract

The residue generated from the processing of *Tacinga inamoena* (cumbeba) fruit pulp represents a large amount of material that is discarded without proper application. Despite that, it is a raw material that is source of ascorbic acid, carotenoids and phenolic compounds, which are valued in nutraceutical diets for allegedly combating free radicals generated in metabolism. This research paper presents a study focused on the mathematical modeling of drying kinetics and the effect of the process on the level of bioactive of cumbeba residue. The experiments of cumbeba residue drying (untreated or whole residue (WR), crushed residue (CR) and residue in the form of foam (FR)) were carried out in a fixed-bed dryer at four air temperatures (50, 60, 70 and 80 °C). Effective water diffusivity (D_eff_) was determined by the inverse method and its dependence on temperature was described by an Arrhenius-type equation. It was observed that, regardless of the type of pretreatment, the increase in air temperature resulted in higher rate of water removal. The Midilli model showed better simulation of cumbeba residue drying kinetics than the other models tested within the experimental temperature range studied. Effective water diffusivity (D_eff_) ranged from 6.4890 to 11.1900 × 10^−6^ m^2^/s, 2.9285 to 12.754 × 10^−9^ m^2^/s and 1.5393 × 10^−8^ to 12.4270 × 10^−6^ m^2^/s with activation energy of 22.3078, 46.7115 and 58.0736 kJ/mol within the temperature range of 50–80 °C obtained for the whole cumbeba, crushed cumbeba and cumbeba residue in the form of foam, respectively. In relation to bioactive compounds, it was observed that for a fixed temperature the whole residue had higher retention of bioactive compounds, especially phenolic compounds, whereas the crushed residue and the residue in the form of foam had intermediate and lower levels, respectively. This study provides evidence that cumbeba residue in its whole form can be used for the recovery of natural antioxidant bioactive compounds, mainly phenolic compounds, with the possibility of application in the food and pharmaceutical industries.

## 1. Introduction

Cumbeba (*Tacinga inamoena*) is a plant of the Cactaceae family native to and typical of northeastern Brazil, which produces fruits called cumbeba, the size of a plum, with orange–yellow peel and yellow fleshy pulp that surrounds the seeds. The fruit has been reported as a source of bioactive compounds, such as phenolic compounds, carotenoids, betalains and vitamin C [1,2,3,4,5,6,7], substances that have been suggested as having potential beneficial effects on human health [8,9,10,11]. Moreover, its exotic nature, the growing recognition of its nutritional and therapeutic value and its sensory characteristics, which are unprecedented in the market, have shown its great potential as a raw material for agroindustry [1,2,4].

The processing of cumbeba results in high percentages of residues (64.70–79.01% of the processed fruit) [1,2,6] that contain antioxidant compounds whose benefits for health have been reiterated, and their disposal, in addition to causing waste of added value, causes environmental impact and imposes extra costs with transport and discharge in landfills. Therefore, the use of cumbeba residue to generate a product with high added value, as has been done with other agro-industrial by-products, for example the extraction of antioxidant bioactive compounds [12,13,14] and natural dyes [15,16,17], among many other materials [18,19,20,21], can lead to economic gains, prevent or reduce environmental problems caused by the direct disposal of waste in the environment and contribute to the development of a more profitable and sustainable cumbeba production chain.

However, for this to happen, an intermediate processing needs to create a stable form of cumbeba residue to facilitate its subsequent management, such as valorization or storage, because its original wet form (82.60 to 89.67%, wet basis (w.b.)) [1,3] requires virtually instantaneous use or, to ensure its preservation, it must be frozen until use, which, in addition to making its transport expensive, generates additional operating costs. On the other hand, dry products have the advantage of being more stable for a longer period of time [22,23] and occupy a smaller volume, thus promoting reduction of costs with packaging, storage and transportation [24,25]. Drying has also been frequently used before the extraction of antioxidant bioactive compounds from agro-industrial by-products [26,27,28] and from other types of plant biomass [29,30,31], aiming not only to promote preservation until use, but also to prevent interference of water in the extraction process, which improves the efficiency and yield of extraction of the target compounds [32,33,34].

Convective drying can be an excellent pretreatment, even better than freeze drying, prior to obtaining extracts rich in bioactive compounds and with high antioxidant capacity [35]. The content of phenolic compounds and antioxidant activity of olive leaf extracts were significantly increased after convective drying of leaves at high temperatures (3 h, 105 °C) [36]. Wei et al. [37] found that the total free radical elimination capacity of bitter gourd extract increased significantly with the increase in the convective drying temperature (40–60 °C), while the contents of phenolic compounds and flavonoids were not affected. This behavior may be due to the fact that changes in cell structure (cell walls, vacuoles, etc.), disruption of bonds with molecules of the plant matrix (carbohydrates and proteins) and/or inactivation of endogenous enzymes (hydrolytic and oxidative), as a result of the drying process [38,39,40], facilitate the extraction of the associated bioactive compounds [41,42].

Convective hot air drying of plant biomass can be simulated with mathematical models, a potentially useful tool to optimize the process, as it helps to understand the mechanisms that affect water transport, whether internal or external, during drying. These models are divided into three categories: theoretical, empirical and semi-empirical [43,44,45]. While theoretical or diffusive models consider not only external conditions (in the form of boundary conditions), but also the mechanism of resistance to water transfer within the product, based mainly on Fick’s second law, empirical and semi-empirical models consider, under isothermal conditions, only external resistance to water transfer between the product and the drying air [46,47,48]. Although different statistical parameters can be used to evaluate the performance of mathematical drying models [49], in general, the model that best describes the drying curve of the product is the one that has the highest values of correlation coefficient, coefficient of determination and/or modeling efficiency and the lowest values of chi-square, mean squared deviation, mean relative percentage error, mean polarization error, standard error, mean absolute error, residual sum of squares, residuals and/or standard deviation [43,46,47]. Numerous studies have investigated the drying kinetics of different agro-industrial by-products to determine the water diffusivity coefficient and the mathematical model, diffusive and/or empirical, which best represents their drying process [50,51,52,53,54,55,56]. However, for cumbeba residue, with the exception of Ferreira et al. [57], there are no studies, at least as far as we know, on its drying behavior at different temperatures and treatments.

During convective drying, differences in the bioactive compounds of the dry product, compared to the fresh one, have been observed [58,59,60]. However, in general, the extent and effect, positive or negative, of these changes depend on factors such as temperature and/or drying time [59,61], type of chemical material or specific compound [62,63,64,65] and pretreatment applied [66,67,68]. Azeeza et al. [69] observed that phenolic content and antioxidant activity of tomato slices increased with increasing drying temperature (50–70 °C), but decreased as a function of time. Romdhane et al. [70] verified that, regardless of temperature (40–60 °C), the convective drying of lemon peels resulted in a significant reduction in the contents of total phenolics and flavonoids, compared to the fresh material. Mphahlele et al. [64], however, observed no significant differences in total phenolic compounds and flavonoids in pomegranate peel dried at different temperatures (40–60 °C). Başlar et al. [71] found that the levels of bioactive compounds (total phenolics, an-thocyanins and flavonoids) and antioxidant activity were higher in pomegranate arils dried at the highest temperature (75 °C). Cruz et al. [53] also measured an increase in the bioactive potential for grape skin as a consequence of convective drying at 70 °C.

Convective drying of biological materials leads to deviations in quantity, associated with the sensitivity of the compound to heating, and different chemical transformations in bioactive compounds; in addition, in some cases, there may be both the degradation and synthesis of new compounds with antioxidant activity [72,73], which may lead to difficulties in interpreting the results. Therefore, understanding how drying conditions influence the final quality of the dry product, compared to the fresh one, becomes indispensable for better understanding the drying process and for future application of cumbeba residue, such as for the extraction of compounds with active ingredients. In this context, this study was conducted to investigate how temperature and type of pretreatment (whole, crushed or foam) influence the drying characteristics of the cumbeba residue, to determine the drying time, to select the best mathematical model for the drying curves, and to calculate the drying rate, effective diffusivity and activation energy. Finally, in order to evaluate the potential use of cumbeba residue for the extraction of bioactive compounds for functional food formulations, the effects of different drying conditions (temperatures and treatments) on the levels of selected bioactive compounds (total phenolics, flavonoids, anthocyanins and betalains) were investigated.

## 2. Materials and Methods

### 2.1. Material

Cumbeba samples were collected from specimens of *Tacinga inamoena* vegetating in a semi-arid climate, around the coordinates 7°45′3″ S and 37°38′20″ W. Fruits at ripe stage, identified by the yellow color of the peel with orange tones, were chosen and transported to the laboratory in expanded polyethylene containers. At reception, they were washed in running water, brushed to remove the lignified hairs present on the peel, and sanitized in chlorine solution diluted in water in the proportion of 100 mg/kg for 20 min, followed by rinsing in drinking water. The fruits were pulped in a rotary batch pulping machine (Laboremus, Brazil), made of stainless steel and equipped with a screen to separate pulp from residues, composed of seeds and peels with remnants of pulp. The residues were homogenized, placed in low-density polyethylene bags and stored in a cold chamber at −18 °C (HVF-301S; Hesstar, Brazil), from which they were gradually removed to conduct the experimental tests. The residue had an average initial moisture content of 4.00 g of water/g of dry matter (79.91%, w.b.), determined gravimetrically by oven drying at 70 °C and pressure ≤ 100 mmHg until reaching constant weight, according to standard method 934.01 of the Association of Official Analytical Chemists (AOAC) [74]. Emustab emulsifier and Liga Neutra stabilizer, both commercial products from the Du Porto^®^ brand, were used as foaming agents. Folin–Ciocalteu reagent, gallic acid and sodium carbonate were acquired from Sigma Aldrich (St. Louis, MO, USA). All chemicals were analytical grade.

### 2.2. Convective Drying

For use in the tests, the samples were thawed in two steps, first at 4 °C for 24 h and then at approximately 25 °C for 2 h. After this time, the material was divided into three subsamples, the first consisting of whole residue (WR), the second of crushed residue (CR) and the third of residue converted into foam (FR). Untreated or whole residue was dried without any pretreatment. Pretreatment of crushing was performed according to a previous study [57]. For foam formation, in order to facilitate the incorporation and homogenization of the foaming and stabilizing agent, the residue was crushed together with distilled water in the proportion of 2:1 (residue:water, m/m) using a food processor (PMP1600P model, Britânia, Joinville, Brazil) at the maximum speed for 5 min, obtaining a paste with peel and seed fragments, in which the emulsifier (Emustab, Du Porto^®^—2.5 g/100 g paste) and stabilizer (Liga Neutra, Du Porto^®^—1.5 g/100 g paste) were incorporated. Then, the mixture was shaken in a mixer (SX15 model, Arno, São Paulo, Brazil) at the maximum speed (rotation speed level: 3) for 15 min to form the foam. In preliminary tests (data not shown), the foam was considered stable for the foam-mat drying process.

The whole, crushed and foam samples were spread in circular trays, 84.45 ± 0.61 mm in diameter, in layers with a uniform thickness of 9.56 ± 0.44 mm, and subjected to drying in a forced air circulation oven (320 model, Fanem, Guarulhos, Brazil) at temperatures of 50, 60, 70 and 80 °C, with air flow parallel to the trays, at a speed of 1.5 m/s. Before putting the samples into the drying chamber, to ensure that the steady-state condition was reached, the temperature of the dryer was adjusted and maintained for 60 min. The drying behavior was determined by weighing the samples before and during the process, recording the mass reductions on a scale with accuracy of 0.01 g (AS5500C model, Marte, Santa Rita do Sapucaí, Brazil), at regular time intervals, around 1.0 min. The drying experiments were carried out until the samples reached constant mass, that is, when there was no variation in the mass for three consecutive measurements, which was assumed as a state of equilibrium. Since the authors believe that the heterogeneity of the residue can influence the results, in order to obtain reproducibility in the experimental findings, all drying experiments were performed in six replicates. The dry residue was removed from the tray and crushed in a food processor (HC31X-Type 2 model, Black Decker, Uberaba, Brazil), at intervals of a few seconds to avoid thermal stress of the material, resulting in a powder with particle size of 0.07–0.84 mm (sifted through a mesh no. 20–200), which was packed in laminated packaging and stored at −18 °C for further analysis.

In the present study, the data of the cumbeba residue mass, at different time intervals, for the different drying conditions (temperature and pretreatment), were converted into moisture content data (d.b.). Then, the dimensionless moisture content (MR) was calculated from the values of the initial moisture content, moisture content at different time intervals and the equilibrium moisture content, according to Equation (1):(1)$$\mathrm{MR}=\frac{M_{t}-M_{e}}{M_{0}-M_{e}}$$
where M_t_, M_0_ and M_e_ are moisture content at a time t (g of water/100 g of dry matter), initial moisture content (g of water/100 g of dry matter) and equilibrium moisture content (g of water/100 g of dry matter), respectively.

The drying rate for each experiment can be obtained through Equation (2) [75,76]:(2)$$\mathrm{Drying}\;\mathrm{rate}=\frac{M_{t}-M_{t+\Delta t}}{\Delta t}$$
where M_t + Δt_, M_t_ and Δt are the moisture content at t + Δt (g water/g dry matter), moisture content at t (g water/g dry matter) and the difference between the current time (t) and the previous time (t_0_) of drying (min), respectively. Drying behavior was determined using a graph of drying rate versus t for all experiments.

### 2.3. Mathematical Modeling of Drying Kinetics

To model the drying kinetics of cumbeba residue, under the different drying conditions, ten thin-layer drying models were chosen and are detailed in Table 1. The models were fitted to the experimental data using the software program Statistica^®^ version 7.0, through nonlinear regression, by the Quasi-Newton method (7.0.61.0, StatSoft Inc., Tulsa, OK, USA).

The quality of fit of each model to the experimental data was verified using the coefficient of determination (R^2^) (Equation (3)), the mean squared deviation (MSD) (Equation (4)) and the chi-square (χ^2^) (Equation (5)) [46,47]. The model with the highest value of R^2^ and the lowest values of MSD and χ^2^ was selected as the best model to describe the drying kinetics of cumbeba residue.
(3)$$R^{2}=\frac{\sum_{i=1}^{N}{\left[{(\mathrm{MR}}_{\exp,i}-{\bar{\mathrm{MR}}}_{\exp,i}{)\;(\mathrm{MR}}_{\mathrm{pred},i}-{\bar{\mathrm{MR}}}_{\mathrm{pred},i})\right]}^{2}}{\sum_{i=1}^{N}{{(\mathrm{MR}}_{\exp,i}-{\bar{\mathrm{MR}}}_{\exp,i})}^{2}\;\sum_{i=1}^{N}{{(\mathrm{MR}}_{\mathrm{pred},i}-{\bar{\mathrm{MR}}}_{\mathrm{pred},i})}^{2}}$$
(4)$$\mathrm{MSD}={\frac{1}{N}\sum_{i=1}^{N}{{(\mathrm{MR}}_{\mathrm{pred},i}\;{\mathrm{MR}}_{\exp,i})}^{2}}^{\frac{1}{2}}$$
(5)$$\chi^{2}=\frac{1}{N-n}\sum_{i=1}^{N}{{(\mathrm{MR}}_{\exp,i}-{\mathrm{MR}}_{\mathrm{pred},i})}^{2}$$
where MR_exp,i_, ${\bar{\mathrm{MR}}}_{\exp,i}$, MR_pred,i_, ${\bar{\mathrm{MR}}}_{\mathrm{pred},i}$, N and n are the experimental dimensionless moisture content, mean of the experimental dimensionless moisture content, dimensionless moisture content predicted by the model, mean of the dimensionless moisture content predicted by the model, number of observations and number of coefficients (constants) of the model, respectively.

### 2.4. Determination of the Effective Water Diffusivity

The analytical solution of Fick’s second law was used to describe the drying of cumbeba residue (whole, crushed and in the form of foam) considering the geometric shape of the samples as approximate of an infinite plate (area >> thickness). This model assumes: (1) constant convective mass transfer coefficient (h) and effective water diffusivity (D_eff_); (2) homogeneous and isotropic material; (3) uniform initial moisture distribution; (4) diffusion as the only mechanism for water transport; (5) negligible variation in sample volume. For a boundary condition of the third type, MR (t) is given by Equation (6) [86,87]:(6)$$\mathrm{MR}(t)=\sum_{n=1}^{\infty }B_{n}\exp\left(-\frac{\mu_{n}{}^{2}}{{\left(L/2\right)}^{2}}D_{\mathrm{eff}}t\right)$$

In Equation (6), the parameter B_n_ is given by Equation (7):(7)$$B_{n}=\frac{{2\mathrm{Bi}}^{2}}{\mu_{n}{}^{2}{(\mathrm{Bi}}^{2}+\mathrm{Bi}+\mu_{n}{}^{2})}$$
where Bi is the mass transfer Biot number:(8)$$\mathrm{Bi}=\frac{\mathrm{hL}/2}{D_{\mathrm{eff}}}$$
where h is the convective mass transfer coefficient (m/s), L is the thickness of infinite plate (m) and D_eff_ is the effective mass diffusivity (water). In Equations (6) and (7), μ_n_ are the roots of the following transcendental equation:(9)$${\cot\mu }_{n}=\frac{\mu_{n}}{\mathrm{Bi}}$$

The roots of Equation (9) were calculated for different values of mass transfer Biot number (Bi) (0 ≤ Bi ≤ 200), with the effective mass diffusivity (D_eff_) and the convective mass transfer coefficient (h) determined by minimizing the chi-square objective function [88,89], according to the optimization algorithm proposed by Silva et al. [90], using 16 terms of the series given in Equation (6), employing the software program Convective Adsorption—Desorption (Federal University of Campina Grande, PB, Brazil).

### 2.5. Determination of Activation Energy

In the present study, an Arrhenius-type equation, Equation (10), was used to relate the effective mass diffusivity (D_eff_) and drying air temperature (T) [25,91,92]:(10)$$D_{\mathrm{eff}}{=D}_{o}\exp\left(\frac{{-E}_{a}}{R(T+273.15)}\right)$$
where D_o_ is the pre-exponential factor (m^2^/s), E_a_ is the activation energy (kJ/mol), R is the universal constant of gases (0.008314 kJ/mol K) and T is the drying air temperature (°C).

### 2.6. Chemical Analyses

#### 2.6.1. Phenolic Compounds

The content of total phenolic compounds (TPC) was determined according to the Folin–Ciocalteu micro-method adapted by Waterhouse [93], which uses the Folin–Ciocalteu reagent, sodium carbonate, methanol and gallic acid to obtain the standard curve. First, 1.0 g of the sample was weighed and 50 mL of distilled water was added, macerating the mixture until complete homogenization. The mixture was then left at rest, at room temperature (25 ± 2 °C) for 30 min, in a dark room; soon after, it was filtered and the Folin–Ciocalteu phenol reagent (125 μL) was added, followed by vigorous shaking and rest for 5 min. After the reaction time, 250 μL of aqueous solution of sodium carbonate (Na_2_CO_3_) (20 g/100 g solution) was added, followed by further shaking and rest in a water bath at 40 °C for 30 min. A reagent blank was prepared using distilled water, Folin–Ciocalteu phenol reagent and sodium carbonate solution. The absorbance of the extract was measured at 765 nm using a UV/Visible spectrophotometer (35-D model, Coleman, Santo André, Brazil). The result was expressed in mg of gallic acid equivalent (GAE)/100 g of dry matter, calculated using Equation (11). The standard curve was obtained by varying the concentration of the gallic acid solution between 0 and 22.5 μg/mL (R^2^ = 0.9994). The analysis was performed in quadruplicate.
(11)$$\mathrm{TPC}=\frac{(A-{b)\times V}_{e}}{{10\times a\times m\times V}_{d}}$$
where A is the measured absorbance, a is the angular coefficient of the equation of the standard curve, b is the linear coefficient of the equation of the standard curve, m is the dry sample mass (g), V_e_ is the volume of the extract (mL) and V_d_ is the volume of dilution (mL).

#### 2.6.2. Flavonoids and Anthocyanins

Contents of total flavonoids (TF) and total anthocyanins (TA) were determined according to the methodology described by Francis [94], where 1.0 g of the sample was weighed, 10 mL of ethanol-HCl solution (1.5 N) in the ratio 85:15 (v:v) was added and then the mixture was macerated for 1 min. The extract was collected in a test tube and kept under refrigeration (5 °C) for 24 h. After this period, the extract was filtered in cotton and reading was performed in UV/Visible spectrophotometer (35-D model, Coleman, Santo André, Brazil) at 374 nm for flavonoids and at 535 nm for anthocyanins. The results were expressed in mg/100 g of dry matter, calculated using the Equations (12) and (13). The analysis was performed in quadruplicate.
(12)$$\mathrm{TF}=\frac{A\times F_{d}}{76.6}$$
(13)$$\mathrm{TA}=\frac{A\times F_{d}}{98.2}$$
where A is the measured absorbance and F_d_ is the dilution factor of the extract, calculated according to Equation (14).
(14)$$F_{d}=\frac{100\times V}{m}$$
where m is the dry sample mass (g) and V the dilution volume (mL).

#### 2.6.3. Betalains

Betalains (betaxanthins and betacyanins) were determined according to Castellar et al. [95] with some adaptations. The extracts were prepared using ethanol:water solution in the ratio of 80:20 (*v*:*v*) as extraction solvent. Approximately 1.0 g of sample was macerated in 10 mL of 80% ethanol, stirred and then stored under refrigeration (5 °C) for 24 h. After this period, the mixture was centrifuged at 4 °C and 3500 rpm for 10 min. Then, the supernatant was collected in a graduated cylinder and the insoluble part was reextracted with more 10 mL of 80% ethanol, which was subjected to the same procedure described above. The two supernatants were combined and the final volume adjusted to 25 mL with 80% ethanol; finally, the extract was again subjected to shaking. The absorbances of the extracts were measured in a UV/Visible spectrophotometer (35-D model, Coleman, Santo André, Brazil) at 480 nm for betaxanthins and at 535 nm for betacyanins. The results were expressed in mg/100 g of dry matter, calculated according to Equation (15). The analysis was performed in quadruplicate.
(15)$$\mathrm{Betax}.\;\mathrm{or}\;\mathrm{Betac}.=\frac{{A\times F}_{d}\times \mathrm{MW}\times V\times 100}{€\times W\times m}$$
where A is absorbance measured at 480 nm (betaxanthins) or 535 nm (betacyanins), F_d_ is the dilution factor of the extract, MW is molecular weight (308 g/mol and 550 g/mol for betaxanthins and betacyanins, respectively), V is the volume of the extract, € is the coefficient of extinction (48,000 L/mol cm and 60,000 L/mol cm for betaxanthins and betacyanins, respectively), W is the width of the spectrophotometer curve (1 cm) and m is the dry mass of the sample (g).

### 2.7. Statistical Analysis

The results were expressed as the mean ± standard deviation and data analysis was performed using Statistica software version 7.0 (7.0.61.0, StatSoft Inc., Tulsa, OK, USA). The differences between treatment means were determined using one-way analysis of variance (ANOVA) and a Tukey multiple comparison test in a confidence interval of 95% (*p* < 0.05).

## 3. Results and Discussion

### 3.1. Drying Kinetics

At the beginning of the process (t = 0 and MR = 1), the amount of water of the untreated or whole residue (WR), crushed residue (CR) and residue in the form of foam (FR) was 4.00 ± 0.34 g of water/g of dry matter (79.91 ± 1.38%, w.b.), 4.03 ± 0.26 g of water/g of dry matter (80.07 ± 1.04%, w.b.) and 5.25 ± 0.34 g of water/g of dry matter (83.95 ± 0.88%, w.b.), respectively.

#### 3.1.1. Influence of Air Temperature

The experimental curves of drying kinetics of WR, CR and FR, at temperatures of 50, 60, 70 and 80 °C, which describe the evolution of the dimensionless moisture content over time, are shown in Figure 1a–c. It can be observed that the dimensionless moisture content (MR) decreased continuously over time and, in addition, with the increase in drying temperature, regardless of the pretreatment applied to the residue, the time required for the samples to reach the equilibrium moisture content became progressively shorter. For WR (Figure 1a), the drying times correspond to 1320, 900, 840 and 600 min, with final moisture contents (MR = 0) of 6.06, 5.99, 10.84 and 9.15% (d.b.), at temperatures of 50, 60, 70 and 80 °C, respectively. For CR (Figure 1b), at the four temperatures, the drying times were 1560, 1320, 1080 and 780 min, with final moisture contents of 6.60, 5.30, 10.43 and 10.10% (d.b.), while for FR (Figure 1c) the drying times were 1380, 840, 480 and 420 min, with final moisture contents of 7.58, 6.31, 11.04 and 9.20% (d.b.), at temperatures of 50, 60, 70 and 80 °C, respectively. It was observed that, with the increase in drying temperature from 50 to 80 °C there were reductions of 54.54% (720 min), 50.0% (780 min) and 69.56% (960 min) in the drying time of WR, CR and FR, respectively. The increase in temperature promotes a higher rate of heat transfer [96], causing a higher degree of agitation of water molecules [97] and, therefore, a higher vapor pressure in the sample [98], which translates into an increase in their mobility [99,100], which can accelerate the removal of water and thus reduce drying time. Similar results have been reported in the literature [101,102,103].

Figure 2a–c show the variation of the drying rate over time, under the different drying conditions, where it is possible to observe that, for all types of pretreatments, the increase in temperature resulted in a higher drying rate. The increase in temperature favors heat transfer, which results in faster heating and greater vibration of water molecules and, therefore, a higher vapor pressure in the sample [96,97,98]. This can accelerate water removal. In addition, it is verified that the drying process of WR, CR and FR occurs in almost three stages. During the initial stage, the first drying period, common to all drying conditions studied, due to the rapid increase in the temperature of the samples (initial heating) [104,105,106,107] there was a higher drying rate at the beginning of the process, with a short acceleration period in which the drying rates gradually increased to the maximum value.

In the second stage, the second drying period, immediately after the peak (maximum drying rate), which appears more clearly at temperatures of 50, 60 and 70 °C, in the residue in the form of foam (Figure 2c) the drying rate decreased sharply and then maintained little variation (stability), revealing that water evaporation at the product–air interface occurs at a rate similar to that of water diffusion from the inside of the product to its surface. The porous structure of the samples on FR can possibly facilitate the internal transfer of moisture, so as to keep the surface saturated for a long time. As drying progresses, in the final drying stage (third period or falling rate period), as the surface of the samples became unsaturated with moisture and the drying front was displaced to their interior, thermal diffusion was increasingly difficult, because the temperature gradient between the surface and the internal layers of the samples is progressively reduced [104,107,108]. Consequently, water, which is mainly located in the internal layers [109,110,111], needs to take an increasingly longer path to the surface [112], where it will be removed by drying air flow, so the drying rate continuously decreased over time until the equilibrium between the samples and drying air was established (drying rate = 0, t = t_∞_). Similar behavior was observed by Cuevas et al. [56] during drying of olive biomass at different temperatures (69.85–119.85 °C).

It is important to highlight that the FR at the temperature of 80 °C (Figure 2c) had only the acceleration period, referred to here as the first drying period, followed by a much longer period of decreasing rate with time (falling rate period), with no period during which the drying rate was constant or tended to stability. The drying behavior of the FR mentioned above can be explained possibly by the fact that, at the temperature of 80 °C, the outer layers were rapidly dried and evaporation was occurring mainly in the internal regions, which cannot provide a stable supply of water to the surface of the foam, characteristic of the second drying period. This phenomenon is the result of the withdrawal of the drying front towards the center of the sample (porous medium), characterized by the loss of connectivity between the exposed surface and the water inside the material [113]. In this stage, the internal resistance to the molecular transport of water (diffusion), which occurs as a result of a gradient of water concentration inside the product [33], becomes much higher than the external resistance to the removal of water vapor on the surface of the samples by the drying air [114,115,116]. Similar results were found in the drying of pumpkin pulp foam [117], yacon (*Smallanthus sonchifolius*) juice foam [118] and tomato pulp foam [119].

#### 3.1.2. Influence of the Type of Pretreatment

Regarding the influence of the type of pretreatment (WR, CR and FR) on MR behavior with the drying time presented in Figure 3a–d, it should be noted that, at temperatures of 50 and 60 °C (Figure 3a,b, respectively), there was no marked difference during the initial period between the drying curves of the residue in the form of foam and crushed, and that the WR took less time (1320 min), at 50 °C, to reach the equilibrium (MR = 0) than CR (1560 min) and FR (1380 min). On the other hand, for drying temperatures of 60, 70 and 80 °C (Figure 3b–d), although the drying curves of FR and WR do not diverge from each other, especially at 70 and 80 °C, the FR showed the shortest drying times (840, 480 and 420 min, respectively), followed by WR (900, 840 and 600 min, respectively) and CR (1320, 1080 and 780 min, respectively). This occurs because the internal structure, more porous and uniform, and the large exposed surface area of FR facilitate the transfer of heat and mass to the drying air [112,120,121], which eventually improves water loss (rate) at the end of the process, compared to WR and CR, mainly at high temperatures, leading to a faster drying process.

The drying rates of cumbeba residue subjected to different pretreatments for a specific temperature are shown in Figure 4a–d. Although WR at temperatures of 50 and 60 °C (Figure 4a,b, respectively), mainly in most of the second drying period (40 min < t ≤ 180 min), showed a slightly higher drying rate (maximum values reached of 0.027 and 0.033 g water/g dry matter.min, respectively), the FR showed higher values during virtually the rest of the drying period (falling rate period). On the other hand, at temperatures of 70 and 80 °C (Figure 4c,d, respectively), FR showed in general the highest drying rates (maximum values reached of 0.032 and 0.042 g water/g dry matter.min, respectively), followed by WR (0.032 and 0.033 g water/g dry matter.min, respectively) and CR (0.021 and 0.029 g water/g dry matter.min, respectively).

The drying behavior (see Figure 4a–d) of the residue subjected to the different pretreatments tested can be explained by the structural characteristics of each type of sample. Compared to CR and FR, WR showed a more heterogeneous but more porous structure, which allowed contact between the drying air and the inner layers of the samples, resulting in better heat and mass transfer, especially at lower temperatures (50–60 °C). However, it was observed that, at high temperatures (70–80 °C), a dry and little permeable crust formed in the fragments of peels linked to the residual pulp, which led to a greater resistance to heat and mass transfer, hence hindering water removal. In addition, the presence of seeds, due to their compacted and fibrous nature, may have been another obstacle to a more efficient water removal. These effects seem to have a smaller relative influence on the transport of water at low temperatures, which may also explain the variation that occurred in the final moisture content of the product (See Section 3.1.1).

In turn, CR, despite being more homogeneous compared to the whole residue, was formed by small fragments of peel, residual pulp and seeds, which were grouped into a compacted, little porous structure, which evidently played a relevant role in the drying process, preventing greater water loss and, as a result, led to lower drying rates. On the other hand, the FR showed, in the initial moments of drying, a type of volumetric expansion (data not shown), indicating the existence of a physical barrier on its surface, possibly formed by small fragments of peel, which hindered the transfer of mass, in the form of vapor, between the FR and the drying air. However, this phenomenon occurred for longer in the drying at temperatures of 50 and 60 °C, while at temperatures of 70 and 80 °C, on the surface of the FR, despite also showing a certain expansion, it was less pronounced and had shorter duration, possibly because the rapid evaporation of water on the surface caused the collapse of its structure, a certain anisotropic contraction (shrinkage) in relation to the wall of the trays which led to the exposure of the inner layers of the FR to the drying air. Apparently, shrinkage due to water loss was more important than the effect of expansion associated with resistance to water vapor loss on the surface of the FR. In any case, these combined phenomena directly led to higher values of drying rate observed. However, for better insights into this subject, knowledge on heat and mass balances on the surface and inside the FR is critical. To this end, more complete experiments can be designed, in which computational modeling can provide essential complementary information.

### 3.2. Mathematical Modeling of the Experimental Data of Drying Kinetics

The moisture content data obtained in the drying process were converted into MR (Equation (1)) and the ten mathematical drying models listed in Table 1 were fitted. Table 2 shows the coefficients of the models and the parameters used to evaluate the quality of fit (R^2^, MSD and χ^2^) to the drying curves of the experimental data of WR, CR and FR, in the experimental air temperature range (50–80 °C). It can be observed in Table 2 that the quality of fit of the models depends on the type of pretreatment applied to the cumbeba residue. For WR and CR, all models had high values of R^2^ (>0.990) and low values of MSD (<0.0344) and χ^2^ (<0.0013). Among the models, Midilli (model 10) and Approximation of Diffusion models (model 7) had the highest values of R^2^ (0.9984–0.9991 and 0.9982–0.9989, respectively) and lowest values of MSD (0.0106–0.0147 and 0.0119–0.0158, respectively) and χ^2^ (0.0001–0.0003 and 0.0002–0.0003, respectively). However, for FR, the Midilli model (model 10) was the only one to have, at all drying temperatures studied, R^2^ values higher than 0.990 with corresponding values of MSD and χ^2^ values lower than 0.0233 and 0.0007, respectively.

Although the adequacy of the Approximation of Diffusion model (model 7) was comparable to that of the Midilli model (model 10), when all drying conditions were analyzed, the latter model was more in agreement with the experimental data. As can be seen in Table 2, the values of R^2^ for the Midilli model were higher than those of the Approximation of Diffusion model, while the values of the other statistical parameters (MSD and χ^2^) were lower. Based on these results, the Midilli model was chosen as the best model to represent the drying of cumbeba residue under the different conditions studied. Thus, the moisture content at any time during the drying process can be reliably estimated using the Midilli model. Similar results have been found in previous studies [122,123,124]. Figure 5a–c show the adequacy of the Midilli model to the experimental data for temperatures ranging from 50 to 80 °C for WR, CR and FR.

The validation of the Midilli model was established by the comparison between the experimental data of the dimensionless moisture content and the data estimated/predicted by the model under any specific drying condition. Figure 6a–c show the dimensionless moisture content predicted by the Midilli model versus the experimental dimensionless moisture content of WR, CR and FR at the different temperatures (50 to 80 °C). As expected, there is an adequate correlation between the experimental and predicted values of MR, since the data are grouped around the straight line, which theoretically represents the equality between experimental and estimated values, evidencing the adequacy of the Midilli model in the description of the drying behavior of WR, CR and FR in the temperature range evaluated.

### 3.3. Effective Water Diffusivity (D_eff_) and Activation Energy (E_a_)

The effective water diffusivity coefficient (D_eff_) of the samples was determined by computer simulation (see Section 2.4), and the results are presented in Table 3. Ferreira et al. [57] proposed a diffusion model, assuming a convective boundary condition, to describe the convective drying of crushed cumbeba residue. The parameters of mass transport (D_eff_ and h) and activation energy (E_a_) determined in Ref. [57] were used in this article. As shown in Table 3, the D_eff_ values of the samples were influenced by the drying temperature and type of pretreatment. The values of D_eff_, for the same type of pretreatment, increased as the drying temperature increased from 50 to 80 °C from 6.49 to 11.19 × 10^−6^ m^2^/s for WR, from 2.93 to 12.75 × 10^−9^ m^2^/s for CR and from 1.54 × 10^−8^ to 12.43 × 10^−6^ m^2^/s for FR. This is due to the fact that increasing the drying air temperature increases the overall temperature of the product [104], resulting in higher kinetic energy of water molecules (agitation) [123,125], consequently facilitating water diffusion towards the external layers of the product [48,126], which causes the increase in D_eff_ values. Comparable results on the change of D_eff_ with temperature have been observed by different authors in the drying of different types of plant biomass [107,126,127,128,129].

Regarding the type of pretreatment, interestingly, it was found that at temperatures of 50 °C and also 70 °C the D_eff_ values of WR were higher, followed by those of FR and CR. On the other hand, at temperatures of 60 and 80 °C, this behavior was reversed, with FR showing the highest values of D_eff_. Although this result was consistent with the fact that, at the temperature of 80 °C, the porous structure of the foam increased the rate of water removal and, as a consequence, reduced the drying time, we believe that the absence of trend in D_eff_ values observed between WR and FR may be attributed to the heterogeneity of cumbeba residue, resulting from the different chemical compositions and uneven distribution of its constituents (peel, residual pulp and seeds) in the drying experiments (data not shown).

The D_eff_ values determined, ranging from 10^−9^ to 10^−5^ m^2^/s, are close to or higher than the range reported by Madamba et al. [130] for fruits, which according to the authors is between 10^−11^ and 10^−9^ m^2^/s. In this case, it is necessary to take into account other factors that equally influence D_eff_, such as shrinkage during drying [131,132], the initial distribution of water in the product, according to Younis et al. [133], and the temperature and/or composition [110,134,135], which in the case of cumbeba peel possibly has a selective permeability to water as a mechanism of adaptation of the species to the semi-arid climate. In addition, this deviation can be attributed to the fact that cumbeba residue, when compared to fruit and vegetable samples, has a more porous structure and a large surface area, which facilitates the transfer of heat and mass in the form of water vapor within the samples. A similar trend was found during the drying of grape residue [136] and the by-product of uvaia (Eugenia pyriformis) pulp processing [60].

It is interesting to mention that the values of convective mass transfer coefficients (h) (see Table 3) also showed, regardless of the pretreatment applied to the cumbeba residue, a trend of increase with the increase in temperature, indicating that the drying air has greater capacity to remove moisture from the surface of the samples as its temperature increases. On the other hand, the low values of Biot number (Bi) (1.00 × 10^−3^–2.10 × 10^0^), obtained under different experimental conditions, indicate that there is resistance to the mass flow (water) on the surface of the samples and therefore show that the solution of the diffusion equation (Equation (6)) considering a third-type boundary condition, even with limitations, as it does not consider the variation in product dimensions and the non-linearities of the thermophysical properties, is reasonably adequate to describe the drying process of cumbeba residue, especially for the whole and crushed residue (R^2^ ≥ 0.9948, see Table 3). The activation energies (E_a_) were determined by fitting Equation (10) by simple nonlinear regression to the ordered pairs (T, D_eff_) for each type of pretreatment (Table 3). As shown in Table 3, the activation energies were determined at 22.31, 46.71 and 58.07 kJ/mol for WR, CR and FR, respectively. FR had the highest value of E_a_, indicating that the effective diffusivity of the foam was more sensitive to the temperature change, compared to WR and CR. The activation energy values found (22.31–58.07 kJ/mol) are within the range of the values reported for plant biomass (12.7–110 kJ/mol) [137].

### 3.4. Effect of Drying Conditions on the Level of Bioactive Compounds of Cumbeba Residue

#### Total Phenolic Compounds (TPC)

The content of phenolic compounds of the residue was significantly (*p* < 0.05) affected by the drying conditions. In general, the content of phenolic compounds varied with the drying condition, in descending order: WR dried at 80 °C > CR dried at 80 °C > WR dried at 70 °C > FR dried at 80 °C > CR dried at 70 °C > WR dried at 60–50 °C > fresh sample > CR dried at 60 °C > FR dried at 70 °C > CR dried at 50 °C > FR dried at 60–50 °C (Table 4). A statistically significant increase (*p*< 0.05) in phenolic content, compared to the fresh sample (436.719 mg GAE/100 g dry matter), can be observed for all treatments, except for CR and FR, where phenolic concentration was only higher than that in the fresh sample from temperatures of 70 and 80 °C, respectively (Table 4).

According to Table 4, as the temperature increased from 50 to 80 °C, there was an increase of 23.65% in the content of phenolic compounds of WR, while increments of 33.24% and 36.21% were recorded for CR and FR, respectively. In the present study, the WR dried at 80 °C allowed a greater recovery of total phenolics (585.171 mg of GAE/100 g of dry matter), corresponding to an increase of about 25.40% in comparison to the fresh residue, which is consistent with observations of previous studies [41,53,71,138]. The increase in phenolic compounds may be possibly due to the release of phenolic compounds from plant cell structures, inactivation of endogenous enzymes (hydrolytic and oxidative) and/or formation of new phenolic compounds [36,39,40,139], as a result of the drying process [38,42,140].

The fresh cumbeba residue had flavonoid and anthocyanin contents of 59.31 mg/100 g of dry matter and 1.44 mg/100 g of dry matter, respectively, while the dry residue had flavonoid content between the limits of 4.16 and 11.16 mg/100 g of dry matter and anthocyanin content between the limits of 0.08 and 0.32 mg/100 g of dry matter (Table 4). The samples showed the following classification for flavonoids, in descending order: fresh sample > WR dried at 80 °C > CR dried at 70 °C > WR dried at 70 °C > CR dried at 80 °C > WR dried at 60 °C > CR dried at 60 °C > WR dried at 50 °C > CR dried at 50 °C > FR dried at 80–50 °C. The anthocyanin content varied, in descending order, as follows: fresh sample > WR dried at 80 °C > WR dried at 60 °C > CR dried at 80 °C > WR dried at 70 °C > CR dried at 70 °C > WR dried at 50 °C = CR dried at 60 °C > FR dried at 80–70 °C > FR dried at 50 °C > CR dried at 50 °C > FR dried at 60 °C.

From the analysis of Table 4, it can be noted that the highest losses of flavonoids and anthocyanins occurred, for all types of sample, compared to the fresh residue, in the drying at 50 °C, indicating that these compounds are sensitive to long exposure to the conditions of the drying process. For example, the percentage reduction values in flavonoid content were 88.56 (1320 min), 85.76 (900 min), 84.02 (840 min) and 81.18% (600 min) (WR), 88.97 (1560 min), 88.37 (1320 min) 83.59 (1080 min) and 84.65% (780 min) (CR) and 92.98 (1380 min), 92.96 (840 min), 92.19 (480 min) and 90.88% (420 min) (FR) at 50, 60, 70 and 80 °C, respectively. Reduction in the contents of these compounds after drying, compared to the fresh material, has also been observed in previous studies [69,70].

Although flavonoids and anthocyanins were the compounds most sensitive to drying, as the temperature increased from 50 to 80 °C, regardless of the type of pretreatment, there was generally a significant retention (*p* < 0.05) of flavonoids and anthocyanins in the samples. While WR resulted in increments of 36.66% and 53.27% in flavonoid and anthocyanin contents, respectively, CR and FR showed increments of 28.10% and 51.17% and of 23.04% and 20.25%, respectively (Table 4). In this context, it seems likely that the reduction of drying time, and possibly of water activity, produced by the use of high drying temperatures reduced the extent of degradation reactions. Furthermore, in the present study, the results suggest that the degradation of total flavonoids and anthocyanins, observed in comparison to the fresh residue, seems to be more dependent on time than on drying air temperature. A possible explanation for this result is that the reduction in flavonoid content may, in certain cases, be due to their involvement in reactions of complexation with other substances and/or to the chemical oxidation of these molecules, which are associated with long periods of exposure to drying air conditions [112,141].

In Table 4, the total betalains content of the fresh residue (18.956 mg/100 g dry matter) is mainly represented by the class of betaxanthins (83.34%), while the remainder (16.66%) is represented by betacyanins (Table 4). The cumbeba residue showed a varying content of betaxanthins, in descending order: fresh sample > WR dried at 80 °C > WR dried at 70 °C > CR dried at 80 °C = FR dried at 80 °C > WR dried at 60 °C > FR dried at 70 °C > CR dried at 70 °C > FR dried at 70 °C > FR dried at 50 °C > WR dried at 50 °C > CR dried at 60 °C > CR dried at 50 °C. On the other hand, the descending order of betacyanin values in dry residue was: fresh sample > CR dried at 80 °C > FR dried at 80 °C > WR dried at 80 °C > CR dried at 70 °C > WR dried at 70 °C > FR dried at 60–70 °C > CR dried at 60 °C > FR dried at 50 °C > WR dried at 60–50 °C > CR dried at 50 °C.

The content of betalains (betaxanthins and betacyanins) of the dry residue was significantly lower (*p* < 0.05) compared to that of the fresh residue (see Table 4). The betaxanthin content was reduced by 60.12–41.02% after drying at 50–80 °C, while a reduction of 60.78–24.12% was observed for betacyanins. This decrease may be due to the decomposition of these antioxidant compounds after being exposed to heating in the drying process [142,143]. However, it is worth pointing out that, for the same type of pretreatment, there was a behavior of retention with the increase in temperature, an effect similar to that observed for the other bioactive compounds analyzed. Although it has been reported that exposure to high temperatures causes the degradation of betalains [143,144], it does not seem to be sufficient to establish a definitive consensus on its sensitivity to changes during drying and how the product matrix may affect this sensitivity. The results seem to support the hypothesis that, as betalains are antioxidant molecules [145,146], prolonged exposure to oxygen seems to be the key factor for their degradation. In addition, the dry WR showed, overall, a content of betaxanthins (6.752–9.317 mg/100 g of dry matter) higher than that of the dry CR (6.300–7.867 mg/100 g dry matter) and dry FR (6.981–7.865 mg/100 g dry matter). As for betacyanins, the dry CR had a higher content (1.239–2.397 mg/100 g of dry matter) compared to the dry FR (1.419–1.901 mg/100 g dry matter) and dry WR (1.268–1.756 mg/100 g dry matter), although these differences were not always significant.

## 4. Conclusions

The drying of cumbeba residues occurred predominantly at a falling rate and the Midilli model was the one that best described the process under all evaluated conditions. The values of effective mass diffusivity and convective mass transfer coefficient increased with increasing drying temperature. Activation energy varied in the following order: residue in the form of foam > crushed residue > whole residue. Convective drying was effective for the recovery of phenolic compounds in cumbeba residues, with better results at 80 °C. Drying of the residue, regardless of pretreatment, led to lower values of flavonoids, anthocyanins and betalains compared to the fresh residue, and this difference was statistically significant. However, the increase of temperature increased the recovery of these bioactive compounds during extraction. Pretreatments of crushing and transformation into foam did not promote an advantage for the extraction of bioactive compounds compared to the untreated residue. Convective drying of whole residue is a good alternative to reuse and add value to cumbeba residue, since it made it possible to obtain a product that can be used as a source of high levels of bioactive compounds, mainly phenolic compounds.

## Figures and Tables

**Figure 1 foods-10-00788-f001:**
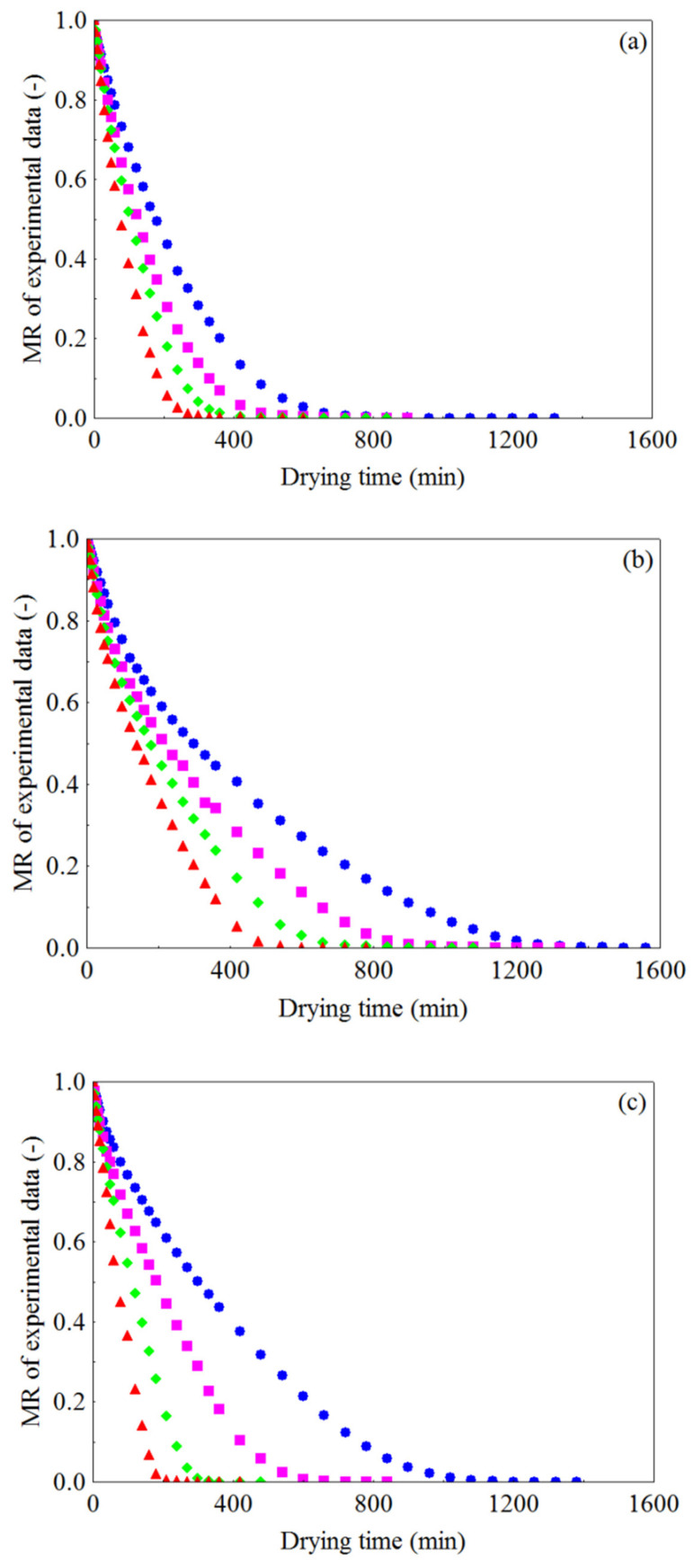
Dimensionless moisture content vs. drying time at different air temperatures for the (**a**) WR (whole residue), (**b**) CR (crushed residue) and (**c**) FR (residue in the form of foam): (
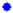
) 50 °C, (
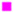
) 60 °C, (
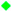
) 70 °C and (
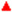
) 80 °C.

**Figure 2 foods-10-00788-f002:**
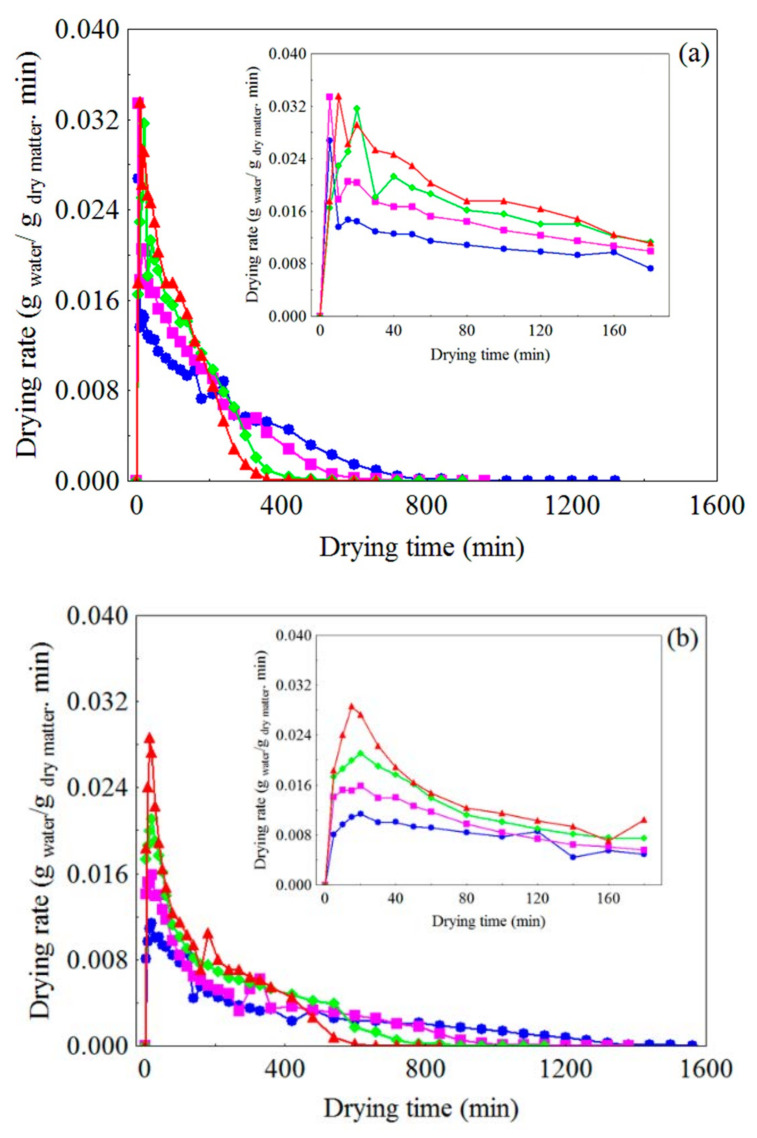
Variation of drying rate over time at different air temperatures for (**a**) WR, (**b**) CR and (**c**) FR: (
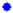
) 50 °C, (
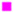
) 60 °C, (
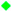
) 70 °C, (
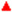
) and 80 °C.

**Figure 3 foods-10-00788-f003:**
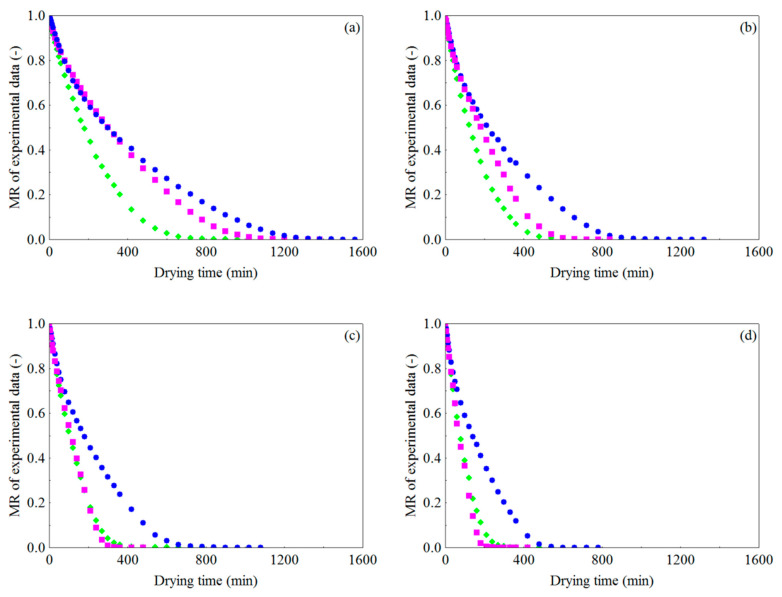
Dimensionless moisture content vs. drying time in different pretreatments for air temperatures of (**a**) 50 °C, (**b**) 60 °C, (**c**) 70 °C and (**d**) 80 °C: (
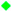
) whole residue, (
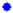
) crushed residue and (
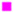
) residue in the form of foam.

**Figure 4 foods-10-00788-f004:**
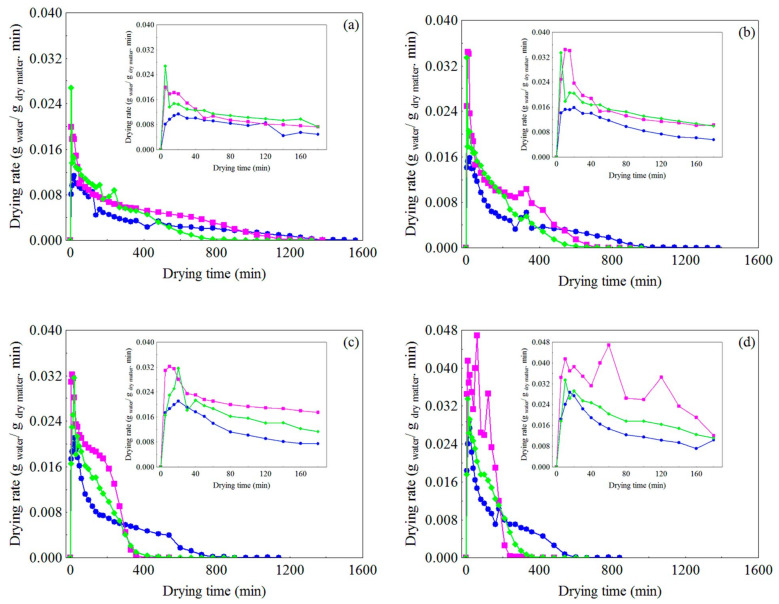
Variation of drying rate vs. drying time in different pretreatments for air temperatures of (**a**) 50 °C, (**b**) 60 °C, (**c**) 70 °C and (**d**) 80 °C: (
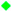
) whole residue, (
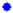
) crushed residue (
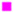
) residue in the form of foam.

**Figure 5 foods-10-00788-f005:**
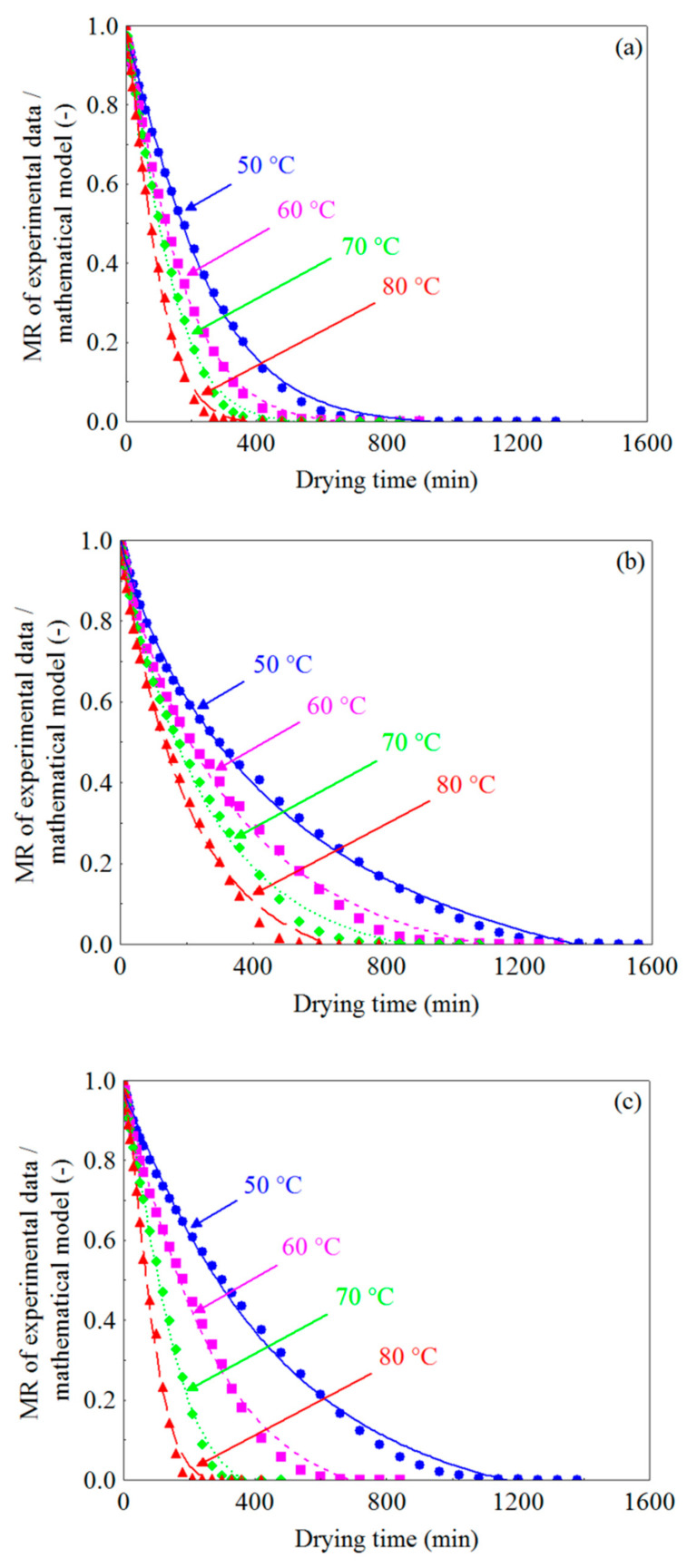
Comparison of dimensionless moisture contents obtained experimentally and predicted using the Midilli mathematical model at temperatures of 50–80 °C for (**a**) WR, (**b**) CR and (**c**) FR: (
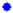
) Experimental data, (

) Model, (
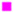
) Experimental data, (

) Model, (
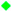
) Experimental data, (

) Model, (
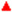
) Experimental data, (

) Model.

**Figure 6 foods-10-00788-f006:**
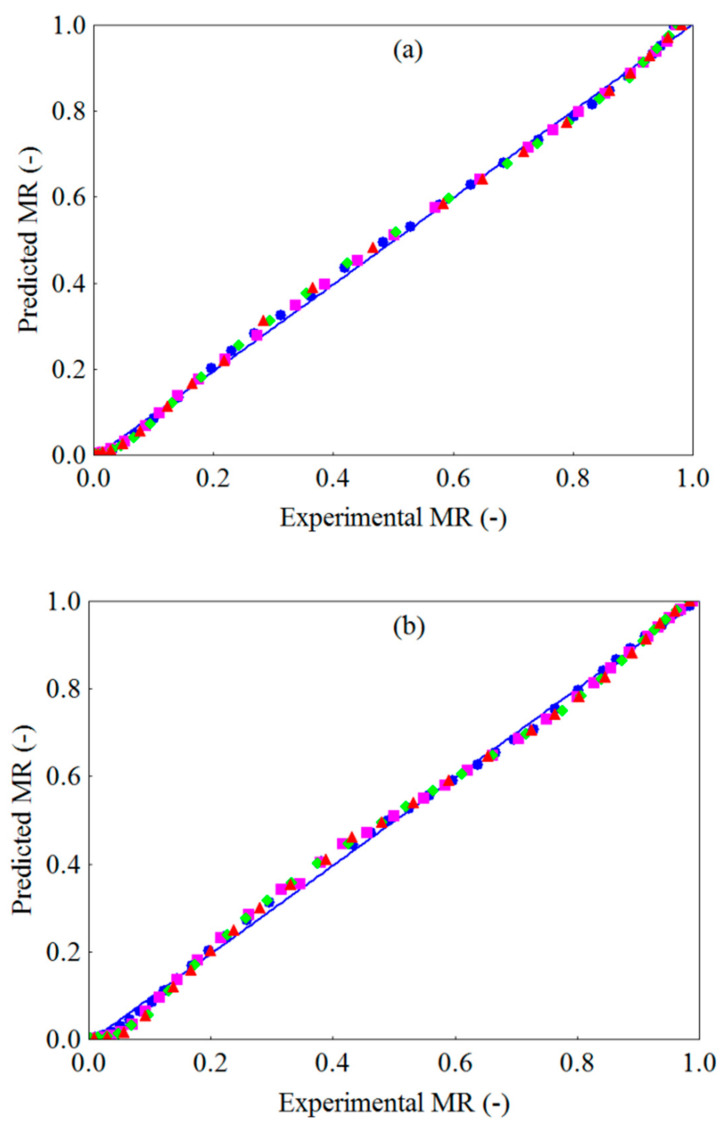
Dimensionless moisture content predicted by the Midilli model versus experimental dimensionless moisture content for (**a**) WR, (**b**) CR and (**c**) FR: (
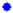
) 50 °C, (
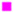
) 60 °C, (
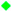
) 70 °C, (
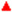
) 80 °C and (

) regression line (MR _Predicted_ = MR _experimental_).

**Table 1 foods-10-00788-t001:** Models fitted to the drying kinetic curves of cumbeba residue.

Model Number	Model Name	Model Equation	References
1	Newton	MR = exp(−kt)	[77]
2	Page	MR = exp(−kt^n^)	[78]
3	Henderson and Pabis	MR = a exp(−kt)	[79]
4	Two-Term Exponential	MR = a exp(−kt) + (1 − a) exp(−kat)	[80]
5	Thompson	MR = exp(−a (a^2^ + 4bt)^0.5^)/2b)	[81]
6	Logarithmic	MR = a exp(−kt) + c	[82]
7	Approximation of Diffusion	MR = a exp(-kt) + (1-a) exp(−kbt)	[80]
8	Modified Henderson and Pabis	MR = a exp(−kt) + b exp(−kt) + c exp(−kt)	[83]
9	Two Terms	MR = a exp(−k_0_t) + b exp(−k_1_t)	[84]
10	Midilli	MR = a exp(−kt^n^) + bt	[85]

MR—Dimensionless moisture content, dimensionless; a, b, c, k, k_0_, k_1_, n—Coefficients of the models; t—Drying time (min).

**Table 2 foods-10-00788-t002:** Results of statistical analyses for models fitted to the convective drying kinetics data of whole residue (WR), crushed residue (CR) and residue in the form of foam (FR).

Model	Treatment	Temp. (°C)	Parameters of Model	R^2^	MSD	χ^2^
1	WR	50	k: 0.2558	0.9958	0.0238	0.0006
		60	k: 0.3638	0.9963	0.0221	0.0005
		70	k: 0.4468	0.9918	0.0334	0.0012
		80	k: 0.6085	0.9913	0.0344	0.0012
	CR	50	k: 0.1437	0.9943	0.0254	0.0007
		60	k: 0.2000	0.9945	0.0261	0.0007
		70	k: 0.2510	0.9946	0.0259	0.0007
		80	k: 0.3241	0.9941	0.0268	0.0007
	FR	50	k: 0.1555	0.9910	0.0339	0.0012
		60	k: 0.2600	0.9895	0.0359	0.0013
		70	k: 0.4415	0.9778	0.0545	0.0031
		80	k: 0.6776	0.9769	0.0579	0.0035
2	WR	50	k: 0.2179; n: 1.1190	0.9980	0.0166	0.0003
		60	k: 0.3249; n: 1.1121	0.9984	0.0144	0.0002
		70	k: 0.3778; n: 1.2049	0.9979	0.0171	0.0003
		80	k: 0.5456; n: 1.2148	0.9983	0.0153	0.0003
	CR	50	k: 0.1520; n: 0.9742	0.9948	0.0253	0.0007
		60	k: 0.2071; n: 0.9787	0.9946	0.0258	0.0007
		70	k: 0.2446; n: 1.0186	0.9947	0.0257	0.0007
		80	k: 0.3142; n: 1.0273	0.9942	0.0264	0.0007
	FR	50	k: 0.1314; n: 1.0897	0.9927	0.0305	0.0010
		60	k: 0.2227; n: 1.1155	0.9922	0.0310	0.0010
		70	k: 0.3417; n: 1.3042	0.9925	0.0318	0.0011
		80	k: 0.5796; n: 1.3925	0.9959	0.0243	0.0006
3	WR	50	a: 0.9835; k: 0.1411	0.9951	0.0245	0.0006
		60	a: 0.9808; k: 0.1947	0.9951	0.0246	0.0006
		70	a: 0.9911; k: 0.2480	0.9948	0.0256	0.0007
		80	a: 0.9941; k: 0.3215	0.9941	0.0266	0.0008
	CR	50	a: 0.9964; k: 0.1548	0.9911	0.0338	0.0012
		60	a: 1.0038; k: 0.2614	0.9896	0.0359	0.0014
		70	a: 1.0425; k: 0.4648	0.9807	0.0509	0.0028
		80	a: 1.0690; k: 0.7341	0.9826	0.0503	0.0028
	FR	50	a: 0.9835; k: 0.1411	0.9951	0.0245	0.0006
		60	a: 0.9808; k: 0.1947	0.9951	0.0246	0.0006
		70	a: 0.9911; k: 0.2480	0.9948	0.0256	0.0007
		80	a: 0.9941; k: 0.3215	0.9941	0.0266	0.0008
4	WR	50	a: 1.6217; k: 0.3307	0.9983	0.0149	0.0002
		60	a: 1.6125; k: 0.4669	0.9988	0.0127	0.0002
		70	a: 1.0000; k: 0.4469	0.9918	0.0334	0.0012
		80	a: 1.7470; k: 0.8477	0.9982	0.0158	0.0003
	CR	50	a: 1.2306; k: 0.1512	0.9946	0.0257	0.0007
		60	a: 1.0000; k: 0.2000	0.9945	0.0261	0.0007
		70	a: 1.0000; k: 0.2510	0.9946	0.0259	0.0007
		80	a: 1.4397; k: 0.3747	0.9947	0.0252	0.0007
	FR	50	a: 1.0000; k: 0.1555	0.9910	0.0339	0.0012
		60	a: 1.6110; k: 0.3335	0.9931	0.0292	0.0004
		70	a: 1.0000; k: 0.4415	0.9778	0.0545	0.0033
		80	a: 1.0000; k: 0.6776	0.9769	0.0579	0.0037
5	WR	50	a: −2782.26; b: 26.6772	0.9958	0.0238	0.0006
		60	a: −2066.58; b: 27.42001	0.9963	0.0221	0.0005
		70	a: −1970.75; b: 29.6745	0.9918	0.0334	0.0012
		80	a: −2031.53; b: 35.1590	0.9913	0.0344	0.0013
	CR	50	a: −1998.18; b: 16.9896	0.9946	0.0257	0.0007
		60	a: −1855.69; b: 19.2688	0.9945	0.0261	0.0007
		70	a: −2174.16; b: 23.3611	0.9946	0.0259	0.0007
		80	a: −2069.70; b: 25.9009	0.9941	0.0268	0.0008
	FR	50	a: −2665.05; b: 20.3600	0.9910	0.0339	0.0012
		60	a: −2196.12; b: 23.8973	0.9895	0.0359	0.0014
		70	a: −2212.75; b: 31.2590	0.9993	0.0545	0.0033
		80	a: −1602.75; b: 32.9552	0.9769	0.0579	0.0037
6	WR	50	a: 1.0335; k: 0.2416; c: −0.0292	0.9975	0.0184	0.0004
		60	a: 1.0370; k: 0.3446; c: −0.0299	0.9979	0.0164	0.0003
		70	a: 1.0600; k: 0.4350; c: −0.0305	0.9952	0.0256	0.0007
		80	a: 1.0691; k: 0.5914; c: −0.0341	0.9956	0.0244	0.0007
	CR	50	a: 1.0224; k: 0.1238; c: −0.0510	0.9970	0.0192	0.0004
		60	a: 1.0125; k: 0.1746; c: −0.0421	0.9970	0.0225	0.0006
		70	a: 1.0259; k: 0.2219; c: −0.0451	0.9970	0.0193	0.0004
		80	a: 1.0380; k: 0.2817; c: −0.0558	0.9969	0.0193	0.0004
	FR	50	a: 1.0800; k: 0.1225; c: −0.1030	0.9974	0.0182	0.0004
		60	a: 1.0874; k: 0.2103; c: −0.1012	0.9959	0.0224	0.0006
		70	a: 1.1617; k: 0.3479; c: −0.1438	0.9916	0.0335	0.0013
		80	a: 1.1189; k: 0.6343; c: −0.0651	0.9880	0.0418	0.0020
7	WR	50	a: −117.6640; k: 0.4010; b: 0.9960	0.9985	0.0141	0.0002
		60	a: −173.6650; k: 0.5634; b: 0.9973	0.9989	0.0119	0.0002
		70	a: −212.5850; k: 0.7682; b: 0.9971	0.9982	0.0158	0.0003
		80	a: −212.4990; k: 1.0585; b: 0.9970	0.9985	0.0145	0.0002
	CR	50	a: −50.6890; k: 0.1641; b: 0.9975	0.9947	0.0256	0.0007
		60	a: −50.0074; k: 0.1390; b: 1.0070	0.9954	0.0240	0.0006
		70	a: −49.9987; k: 0.1633; b: 1.0084	0.9965	0.0209	0.0005
		80	a: −48.9982; k: 0.2039; b: 1.0093	0.9964	0.0181	0.0005
	FR	50	a: −216.9570; k: 0.2327; b: 0.9980	0.9939	0.0279	0.0008
		60	a: −216.0220; k: 0.1427; b: 1.0029	0.9962	0.0217	0.0005
		70	a: −216.0500; k: 0.2060; b: 1.0038	0.9928	0.0310	0.0011
		80	a: −215.9990; k: 0.3670; b: 1.0029	0.9879	0.0419	0.0020
8	WR	50	a: 0.3371; k: 0.2596; b: 0.3371; c: 0.3371	0.9960	0.0233	0.0006
		60	a: 0.3381; k: 0.3706; b: 0.3381; c: 0.3381	0.9965	0.0212	0.0005
		70	a: 0.3455; k: 0.4676; b: 0.3455; c: 0.3455	0.9935	0.0298	0.0010
		80	a: 0.3480; k: 0.6422; b: 0.3480; c: 0.3480	0.9936	0.0295	0.0010
	CR	50	a: 0.3278; k: 0.1411; b: 0.3278; c: 0.3278	0.9975	0.0245	0.0007
		60	a: 0.3269; k: 0.1947; b: 0.3269; c: 0.3269	0.9951	0.0246	0.0007
		70	a: 0.3304; k: 0.2479; b: 0.3304; c: 0.3304	0.9948	0.0256	0.0007
		80	a: 0.3314; k: 0.3215; b: 0.3313; c: 0.3314	0.9941	0.0266	0.0008
	FR	50	a: 0.3321; k: 0.1548; b: 0.3321; c: 0.3321	0.9911	0.0338	0.0013
		60	a: 0.3346; k: 0.2614; b: 0.3346; c: 0.3346	0.9896	0.0359	0.0015
		70	a: 0.3475; k: 0.4648; b: 0.3475; c: 0.3475	0.9807	0.0509	0.0031
		80	a: 0.3563; k: 0.7342; b: 0.3563; c: 0.3563	0.9826	0.0503	0.0031
9	WR	50	a: 0.5056; k0: 0.2596; b: 0.5056; k1: 0.2596	0.9960	0.0233	0.0006
		60	a: 0.5071; k0: 0.3706; b: 0.5071; k1: 0.3706	0.9965	0.0212	0.0005
		70	a: 0.5183; k0: 0.4676; b: 0.5183; k1: 0.4676	0.9935	0.0298	0.0010
		80	a: 0.5219; k0: 0.6422; b: 0.5219; k1: 0.6422	0.9936	0.0295	0.0010
	CR	50	a: 0.49176; k0: 0.1411; b: 0.4917; k1: 0.141131	0.9951	0.0245	0.0007
		60	a: 0.4904; k0: 0.1947; b: 0.4903; k1: 0.194689	0.9951	0.0246	0.0007
		70	a: 0.4955; k0: 0.2480; b: 0.4955; k1: 0.2480	0.9948	0.0256	0.0007
		80	a: 0.4970; k0: 0.3215; b: 0.4970; k1: 0.3215	0.9941	0.0265	0.0008
	FR	50	a: 0.4982; k0: 0.1548; b: 0.4982; k1: 0.1548	0.9911	0.0338	0.0013
		60	a: 0.5019; k0: 0.2614; b: 0.5019; k1: 0.2614	0.9896	0.0355	0.0015
		70	a: 0.5212; k0: 0.4648; b: 0.5212; k1: 0.4648	0.9807	0.0509	0.0031
		80	a: 0.5345; k0: 0.7342; b: 0.5345; k1: 0.7342	0.9826	0.0503	0.0031
10	WR	50	a: 0.9679; k: 0.1876; n: 1.1890; b: −0.0004	0.9989	0.0119	0.0002
		60	a: 0.9734; k: 0.2936; n: 1.1659; b: −0.0007	0.9991	0.0106	0.0001
		70	a: 0.9746; k: 0.3442; n: 1.2664; b: −0.0006	0.9984	0.0147	0.0003
		80	a: 0.9812; k: 0.5169; n: 1.2536; b: −0.0009	0.9987	0.0136	0.0002
	CR	50	a: 1.0070; k: 0.1739; n: 0.8473; b: −0.0037	0.9985	0.0136	0.0002
		60	a: 0.9898; k: 0.2084; n: 0.9250; b: −0.0026	0.9973	0.0184	0.0004
		70	a: 0.9830; k: 0.2338; n: 1.0008; b: −0.0024	0.9967	0.0202	0.0005
		80	a: 0.9850; k: 0.3004; n: 0.9988; b: −0.0040	0.9966	0.0202	0.0005
	FR	50	a: 0.9658; k: 0.1170; n: 1.0785; b: −0.0028	0.9972	0.0188	0.0004
		60	a: 0.9632; k: 0.1908; n: 1.1444; b: −0.0037	0.9962	0.0218	0.0005
		70	a: 0.9575; k: 0.2840; n: 1.3916; b: −0.0047	0.9960	0.0233	0.0007
		80	a: 0.9679; k: 0.5265; n: 1.4834; b: −0.0017	0.9970	0.0205	0.0005

**Table 3 foods-10-00788-t003:** Effective water diffusivity (D_eff_), convective mass transfer coefficient (h), Biot number (Bi) and activation energy for different drying conditions of whole residue (WR), crushed residue (CR) and residue in the form of foam (FR).

Treatment	Temp. (°C)	D_eff_ (m^2^/s)	h (m/s)	Bi (Dimensionless)	R^2^ (Dimensionless)	χ^2^ × 10^−2^ (Dimensionless)	E_a_ (kJ/mol)	R^2^ (Dimensionless)
WR	50	6.49 × 10^−6^	6.79 × 10^−7^	1.00 × 10^−3^	0.9970	2.09	22.31	0.9064
	60	5.57 × 10^−6^	1.02 × 10^−6^	1.75 × 10^−3^	0.9969	1.39		
	70	8.22 × 10^−6^	1.29 × 10^−6^	1.50 × 10^−3^	0.9948	2.41		
	80	1.12 × 10^−5^	1.76 × 10^−6^	1.50 × 10^−3^	0.9955	1.98		
CR *	50	2.93 × 10^−9^	6.44 × 10^−7^	2.10 × 10^0^	0.9962	2.28	46.71	0.9930
	60	4.17 × 10^−9^	8.73 × 10^−7^	2.00 × 10^0^	0.9962	2.12		
	70	8.14 × 10^−9^	8.94 × 10^−7^	1.05 × 10^0^	0.9957	2.14		
	80	1.28 × 10^−8^	10.91 × 10^−7^	8.20 × 10^−1^	0.9951	1.97		
FR	50	1.54 × 10^−8^	4.51 × 10^−7^	2.80 × 10^−1^	0.9927	4.35	58.07	0.8458
	60	6.60 × 10^−6^	6.90 × 10^−7^	1.00 × 10^−3^	0.9916	3.74		
	70	3.25 × 10^−6^	1.19 × 10^−6^	3.50 × 10^−3^	0.9863	6.54		
	80	1.24 × 10^−5^	1.95 × 10^−6^	1.50 × 10^−3^	0.9878	5.60		

* Values determined by Ferreira et al. [57] for drying of the crushed residue were used in this article.

**Table 4 foods-10-00788-t004:** Bioactive compounds of cumbeba residue both fresh and dried under different conditions.

Drying Conditions		TPC	TF	TA	TB	
					Betax.	Betac.
Fresh samples		436.71 ± 2.68 ^h^	59.31 ± 0.03 ^a^	1.44 ± 0.03 ^a^	15.79 ± 0.22 ^a^	3.16 ± 0.15 ^a^
WR	50	446.78 ± 0.55 ^g^	6.78 ± 0.01 ^h^	0.15 ± 0.011 ^f,g^	6.75 ± 0.06 ^g^	1.27 ± 0.00 ^h,i^
	60	485.04 ± 0.97 ^f^	8.44 ± 0.01 ^f^	0.28 ± 0.00 ^c^	7.25 ± 0.02 ^e^	1.34 ± 0.03 ^h,i^
	70	535.98 ± 1.01 ^c^	9.47 ± 0.04 ^d^	0.23 ± 0.01 ^d,e^	8.32 ± 0.02 ^c^	1.62 ± 0.00 ^d,f,e^
	80	585.17 ± 0.99 ^a^	11.16 ± 0.01 ^b^	0.32 ± 0.01 ^b^	9.32 ± 0.02 ^b^	1.75 ± 0.03 ^c,d^
CR	50	370.11 ± 0.96 ^k^	6.54 ± 0.01 ^i^	0.12 ± 0.01 ^g^	6.30 ± 0.02 ^h^	1.23 ± 0.03 ^i^
	60	422.85 ± 0.55 ^i^	6.89 ± 0.01 ^g^	0.15 ± 0.01 ^f,g^	6.72 ± 0.04 ^g^	1.44 ± 0.10 ^e,g,h^
	70	488.84 ± 1.00 ^e^	9.73 ± 0.01 ^c^	0.20 ± 0.01 ^e^	7.11 ± 0.02 ^e,f^	1.67 ± 0.00 ^d,f^
	80	554.41 ± 0.58 ^b^	9.10 ± 0.01 ^e^	0.25 ± 0.01 ^c,d^	7.86 ± 0.04 ^d^	2.39 ± 0.00 ^b^
FR	50	325.57 ± 2.04 ^m^	4.16 ± 0.01 ^l^	0.13 ± 0.00 ^g^	6.98 ± 0.02 ^f^	1.42 ± 0.03 ^g,h,i^
	60	347.19 ± 1.12 ^l^	4.17 ± 0.01 ^l^	0.12 ± 0.00 ^g^	7.10 ± 0.02 ^e,f^	1.57 ± 0.03 ^d,f,e,g^
	70	417.94 ± 1.54 ^j^	4.63 ± 0.01 ^k^	0.14 ± 0.01f ^g^	7.23 ± 0.10 ^e^	1.56 ± 0.07 ^f,e,g^
	80	510.41 ± 0.58 ^d^	5.41 ± 0.01 ^j^	0.16 ± 0.01 ^f^	7.86 ± 0.02 ^d^	1.90 ± 0.03 ^c^

WR—untreated or whole residue; CR—crushed residue; FR—residue in the form of foam; TPC—total phenolic compounds (mg of GAE/100 g dry basis); TF—total flavonoids (mg/100 g dry basis); TA—total anthocyanins (mg 100/g dry basis); TB—total betalains (betaxanthins + betacyanins) (mg/100 g dry basis); Betax.—betaxanthins (mg/100 g dry basis); Betac.—betacyanins (mg/100 g dry basis). The values are means ± standard deviation of the determination in quadruplicate. Means with the same letter in the same column do not differ significantly using a Tukey test (*p* < 0.05).

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
