# Peer review of "Utilization of Cumbeba (Tacinga inamoena) Residue: Drying Kinetics and Effect of Process Conditions on Antioxidant Bioactive Compounds"

_foods, 2021, doi:10.3390/foods10040788_

Round 1

Reviewer 1 Report

Presented manuscript is generally well written and contains some useful information about cumbeba residue drying. However, I have some comments.

Line 337-38 and in the whole manuscript. What is the sense to include four places after comma in effective water diffusivity and activation energy. What were the errors of measurement or calculation of these indices. Two places are probably enough.

Authors  blew up both literature review and results too much. 146 position of literature is rather adequate for review paper.

Figures 1, 3, and 5 present the same results. Only Fig. 5 should be presented. Moreover, authors stated that performed drying experiment in 6 repetition thus standard deviations should be included in charts.  Similarly, figures 2 and 4 present the same result but in different layout.

Conclusions should be more concise. For example first sentence can be deleted. Second sentence is obvious and can be drawn without this study. The same concerning water diffusivity and mass transfer.  

Author Response

Response to Reviewer #1 Comments

Dear reviewer #1

We would like to thank you for your thoughtful comments, improving our manuscript.

Note: Besides the alterations explicitly mentioned, other modifications have been made (in red). The line numbers provided are referring to docx file with the new version of the article.

Presented manuscript is generally well written and contains some useful information about cumbeba residue drying. However, I have some comments.

Line 337-38 and in the whole manuscript. What is the sense to include four places after comma in effective water diffusivity and activation energy. What were the errors of measurement or calculation of these indices. Two places are probably enough.

Authors blew up both literature review and results too much. 146 position of literature is rather adequate for review paper.

Answers: We thank reviewer #1 and agree with his opinion. We have accepted two places after comma for water effective diffusivity as well as activation energy, and throughout the manuscript when possible.

Regarding “...blew up both literature review and results too much”, although it was not our main goal, we wanted this manuscript to also have an aspect of review. However, we can improve the number of positions in the References section if reviewer #1 considers it to be mandatory.

Figures 1, 3, and 5 present the same results. Only Fig. 5 should be presented. Moreover, authors stated that performed drying experiment in 6 repetition thus standard deviations should be included in charts.  Similarly, figures 2 and 4 present the same result but in different layout.

Answer: Because of the way we have written our manuscript, we would not be able to express our results in only one figure. For instance, while Figures 1-2 and 3-4 show the influence of temperature and type of pretreatment on the drying kinetics and drying rate of cumbeba residue, respectively, Figure 5 highlights the fit of Midilli model [85] (best model) to the experimental data of drying kinetics. Therefore, we kindly ask reviewer #1 to keep both the number and the layout of the figures.

Regarding “... performed drying experiment in 6 repetition…”, for each experiment, 6 repetitions were performed. Thus, only the average values were presented in our results. In addition, some programs used in data analyses do not provide statistical tests to show differences between results. We hope that this fact is not significant in the decision of reviewer #1 to recommend publication of our article.

Conclusions should be more concise. For example first sentence can be deleted. Second sentence is obvious and can be drawn without this study. The same concerning water diffusivity and mass transfer.

Answer: We tried to improve the Conclusions section.

Sincerely,

Reviewer 2 Report

The paper “Utilization of cumbeba (Tacinga inamoena) residue: Drying kinetics and effect of process conditions on antioxidant bioactive compounds” by Ferreira et al. deals with the drying of cumbeba residue at different temperature values and after different pretreatments (crushing or foaming). Experimental data were fitted to different possible mathematical models, and values of effective diffusion coefficient and activation energy were also extracted. The retention of bioactive compounds after drying was also monitored.

The study was carefully performed, and the manuscript is well written. But first, the following minor revisions have been taken into account:

line 210: please check the use of subscripts for Mt, M0 and Me

Eq. 4: MDS should actually be MSD

line 248: please check the use of subscript for Bn

line 253: “The first 16 roots of Equation 10 […]” the authors probably wanted to refer to Eq. 9

line 260: “[…] using 16 terms of the series given in Equation (3) […]” the authors probably wanted to refer to Eq. 6?

Eq. 10: shouldn’t it be: D0 exp (-Ea/RT)? Maybe a minus sign is missing?

Line 332: define WR, CR and FR (the reader can guess they refer to whole, crushed and foam residue, but it is better to write it down more clearly the first time these abbreviations are used)

Please fit the three panels of Figure 1 into one page only. Same for Figure 2.

Line 348: “[…] 10.431 10.095% (d.b.) […]” an ‘and’ is missing here

line 521: please check the use of subscript for Deff

Table 3: I cannot understand the meaning of the footnote “Ferreira et al., 2020”

line 563: “[…] solution of the diffusion equation (Equation (7)) […] “ maybe the authors wanted to refer to Eq. 6?

line 568: “[…] by fitting Equation 12 […] “ The authors probably wanted to refer to Eq. 10

Please fit Table 4 into one page only.

Line 615: “For example, the reductions flavonoid […]” please, consider writing something like: “For example, the percentage reduction values in flavonoid content were […]”

line 617: “(1320 m […]” should be “(1320 min), ”

lines 677-678: “Activation energy varied in the following order: residue in the form of foam > whole residue > crushed residue.” this is actually not true. The activation energy of crushed residue is higher than for the whole residue (Table 3).

Author Response

Response to Reviewer 2 Comments

Dear reviewer #2

We would like to thank you for your thoughtful comments, improving our manuscript.

Note: Besides the alterations explicitly mentioned herein, other modifications have been made (in red). The line numbers provided are referring to docx file with the new version of the article.

The paper “Utilization of cumbeba (Tacinga inamoena) residue: Drying kinetics and effect of process conditions on antioxidant bioactive compounds” by Ferreira et al. deals with the drying of cumbeba residue at different temperature values and after different pretreatments (crushing or foaming). Experimental data were fitted to different possible mathematical models, and values of effective diffusion coefficient and activation energy were also extracted. The retention of bioactive compounds after drying was also monitored.

The study was carefully performed, and the manuscript is well written. But first, the following minor revisions have been taken into account:

line 210: please check the use of subscripts for Mt, M0 and Me

Answer: Firstly, thanks for your positive comments (and initial scores). The subscripts have been corrected (Mt, M0 and Me) (line 203).

Eq. 4: MDS should actually be MSD

Answer: Equation (4) has been corrected.

line 248: please check the use of subscript for Bn

Answer: The subscript has been corrected (Bn) (line 241).

line 253: “The first 16 roots of Equation 10 […]” the authors probably wanted to refer to Eq. 9

line 260: “[…] using 16 terms of the series given in Equation (3) […]” the authors probably wanted to refer to Eq. 6?

Answer: Thank you for your observation. Both the text and the numbers of equations have been corrected (please see lines 246-251).

Eq. 10: shouldn’t it be: D0 exp (-Ea/RT)? Maybe a minus sign is missing?

Answer: We agree with the observation of Reviewer #2. Equation (10) has been corrected.

Line 332: define WR, CR and FR (the reader can guess they refer to whole, crushed and foam residue, but it is better to write it down more clearly the first time these abbreviations are used)

Answer: The abbreviations WR, CR and FR have been better defined in the text (lines 322-326).

Please fit the three panels of Figure 1 into one page only. Same for Figure 2.

Answer: Figures 1 and 2 have been readjusted. We hope they are better now.

Line 348: “[…] 10.431 10.095% (d.b.) […]” an ‘and’ is missing here

Answer: Thank you for your observation. The text has been corrected (line 338-339).

line 521: please check the use of subscript for Deff

Answer: The subscript has been corrected (Deff) (please see line 522).

Table 3: I cannot understand the meaning of the footnote “Ferreira et al., 2020”

Answer: We have tried to correct the text and we hope the manuscript is better now (please see lines 518-522 and Table 3).

line 563: “[…] solution of the diffusion equation (Equation (7)) […] “ maybe the authors wanted to refer to Eq. 6?

line 568: “[…] by fitting Equation 12 […] “ The authors probably wanted to refer to Eq. 10

 Answer: We agree with the observation of Reviewer#2. The numbers of equations have been corrected (line 565 and line 569).

Please fit Table 4 into one page only.

Answer: Table 4 has been rearranged and we hope it is better now.

Line 615: “For example, the reductions flavonoid […]” please, consider writing something like: “For example, the percentage reduction values in flavonoid content were […]”

line 617: “(1320 m […]” should be “(1320 min), ”

Answer: The text has been corrected (lines 618-622).

lines 677-678: “Activation energy varied in the following order: residue in the form of foam > whole residue > crushed residue.” this is actually not true. The activation energy of crushed residue is higher than for the whole residue (Table 3).

Answer: We have corrected it and tried to improve the Conclusions section.

Sincerely,

Round 2

Reviewer 1 Report

The authors partially take into consideration my comments.  In my opinion, this paper should be significantly shortened. For example, the same data are presented in charts doubly. However, in Foods journal allows to use of such practices general, this paper can be published in the present form.

Minor

The abbreviation w.b. shuld be explained before the first time using.

Author Response

Response to Reviewer #1 Comments

Dear reviewer #1

We would like to thank you for your thoughtful comments, improving our manuscript.

Note: The line numbers provided are referring to docx file with the new version of the article.

The authors partially take into consideration my comments.  In my opinion, this paper should be significantly shortened. For example, the same data are presented in charts doubly. However, in Foods journal allows to use of such practices general, this paper can be published in the present form.

Minor

The abbreviation w.b. shuld be explained before the first time using.

Answer: Firstly, thanks for your comments. However, because of the way we have written our manuscript, we prefer to present our results as they are.

Regarding “…abbreviation w.b. shuld be explained before…”, the abbreviation w.b. was better defined in the text (line 71).

Sincerely,
